# Care Staff’s Daily Living Decision-Making Support Scale for Older Adults with Dementia in Japan: Development of Validity and Reliability

**DOI:** 10.3390/ijerph192013664

**Published:** 2022-10-21

**Authors:** Mizue Suzuki, Takuya Kanamori, Yukio Koide, Yatami Asai, Masako Sato, Tomoyoshi Naito, Keigo Inagaki, Masao Kanamori

**Affiliations:** 1Faculty of Nursing, Hamamatsu University School of Medicine, 1-20-1 Handayama, Higashi-ku, Hamamatsu 431-3192, Japan; 2Nishiyama Hospital Group, Hamamatsu 431-3192, Japan; 3Geriatric Health Services Facility Mikatahara Bethel Home, Hamamatsu 431-3104, Japan; 4Seirei Mikatahara General Hospital, Hamamatsu 433-8558, Japan; 5College of Sport and Health Sciencies, Ritsumeikan University, Kusatsu 525-8577, Japan

**Keywords:** caregivers, decision-making, dementia, geriatrics, surveys, questionnaires

## Abstract

This study aimed to develop and validate a scale to assess the daily-living decision-making support of care staff for older adults with dementia (OwDs) in Japan. A questionnaire survey was conducted among 138 care staff at two geriatric healthcare facilities from February to March 2021. The Daily Living Decision-Making Support Scale for Older Adults with Dementia (DL-DM) was developed using item analysis, factor analysis, and covariance structure analysis. The factor analysis yielded 12 items and three factors: (1) support for the formation and expression of intentions in daily life based on the life background and values of OwDs; (2) attitudes and devising ways to communicate regarding the formation and expression of intentions in OwDs daily lives; and (3) devising ways to support OwDs in realizing their intentions in daily life. The internal consistency reliability analysis yielded a Cronbach’s α of 0.87 for the total scale. The DL-DM correlated with the concurrent validity measures as expected. The DL-DM demonstrated validity and reliability as a potential scale to assess support for OwDs in daily life decision-making. The results also suggest an association between the DL-DM and person-centered care for OwDs.

## 1. Introduction

The World Health Organization (WHO) defined dementia as “a syndrome—usually of a chronic or progressive nature—that leads to deterioration in cognitive function beyond what affects memory, thinking, orientation, comprehension, calculation, learning capacity, language, and judgement” [1]. Dementia not only causes cognitive decline but also impairs physical function, leaving older adults in need of nursing care or bedridden.

The older adult population is increasing worldwide, and around 55 million people suffer from dementia. This number is expected to rise to 139 million by 2050 [1]. The aging rate in Japan was 29.1% in 2021 and is estimated to reach 39.9% by 2060, indicating that Japan has a significantly larger ultra-aging population than other countries [1]. Additionally, the number of older adults with dementia (OwDs) is increasing. The Ministry of Health, Labour, and Welfare estimated that there were 4.62 million OwDs in June 2013 in Japan [2]. As aging progresses, OwDs are more likely to experience communication problems. Therefore, decision-making support for OwDs, especially support with daily living decision-making, is necessary.

In the “Global action plan on the public health response to dementia,” the WHO promoted awareness about advanced care planning (ACP) for all OwDs to document their wishes for the end of their lives to improve the quality of care toward the end of life [3]. ACP in medical care has been attracting attention because the Japanese Ministry of Health, Labour, and Welfare (MHLW) recommended ACP in its “Guidelines of the decision process of medicine and care in the final stage of life” (re-revised edition) in 2018 [4]. ACP is a process of spontaneous and repeated dialogue with older adults, their family members, and specialists regarding medical treatment and nursing care, not only during treatment of the physical disease but also during the early stages of caregiving to the end of life. The MHLW also published “Decision-making support guidelines in daily life and social life of people with dementia” in 2018, facilitating the recognition of ACP in daily living as one of the most important tasks that can affect the dignity of OwDs [5].

Based on the premise that OwDs still have the ability to make decisions regarding their daily lives, it is recommended that precise support be provided to OwDs to elicit their remaining abilities and capabilities. However, family members or health care staff often make decisions on behalf of OwDs as they believe that OwDs are unable to make appropriate daily-life decisions [6]. In other words, OwDs are suppressed in decision-making because they do not receive adequate support in making decisions in their daily lives.

Few studies have reported on daily-living decision-making support for OwDs, even though assessing decision support in the daily lives of OwDs would facilitate ACP. Although there is a scale to measure nurses’ attitudes toward support for decision-making by OwDs [7], a scale measuring the experience of decision-making support in care staff, including nurses, caregivers, and therapists, toward support for OwDs is necessary. Thus, the purpose of this study was to develop and validate a scale to assess the daily-living decision-making support of care staff for OwDs.

## 2. Material and Methods

### 2.1. Development of the Daily-Living Decision-Making Support Scale for Older Adults with Dementia

To create items for a scale to assess daily-living decision-making support in the care staff of OwDs, three focus group interviews were conducted, with six people in each group. The group interviews were held in September 2020 to gather information on the concrete care methods of decision-making support and ACP that are used by care staff at geriatric care facilities [8]. Focus group interviews were conducted with care staff, such as caregivers, nurses, physical therapists and occupational therapists, at a geriatric care facility. Results were analyzed using a qualitative analysis method called content analysis. After conducting a qualitative analysis of the focus group interviews, items were created to develop the Daily Living Decision-Making Support Scale for Older Adults with Dementia (DL-DM) from October through December, 2020. From January to February 2021, experts on dementia care, including caregivers, physicians, and nurses, were asked to examine the items to determine the items for the final version of the scale. Item responses on the DL-DM are provided on a 4-point Likert scale, where 1 = “never”, 2 = “not very often”, 3 = “sometimes”, and 4 = “most of the time”. Higher scores indicate better practice of decision-making support for OwDs.

### 2.2. Participants

The questionnaire survey was administered to care staff, including nurses, caregivers, physical therapists, occupational therapists, and counselors, at two geriatric care facilities between February and March 2021. Participants were asked to provide information on their age, years of experience in geriatric care, gender, type of job, whether they had experienced an end-of-life care conference (yes/no), experienced asking OwDs on ACP (yes/no), been a person-in-charge of a care meeting (yes/no), participated in a decision-making and ACP workshop (yes/no), and participated in a workshop on person-centered care (yes/no).

The inclusion criterion for the participants was working as care staff in geriatric care facilities, while the exclusion criteria were not being able to give consent, working part-time, and being on maternity leave or other extended leaves of absence. This study comprised 138 participants, and the mean age was 41.46 ± 2.3.

According to Floyd, for a sample size of 100 participants, five participants per variable would be considered adequate to yield reliable results, while 10 participants per variable would be sufficient for a sample size of less than 100 participants [9]. Thus, a sample size of 138 participants in this study was sufficient for factor analysis and covariance structure analysis.

### 2.3. Evaluation of Concurrent Validity

#### 2.3.1. Nurses’ Attitudes toward Support for Decision-Making by OwDs

This scale was used to assess attitudes toward support for decision-making by OwDs. It comprises five subscales: working as a team; respecting the individual’s wishes; trying to understand the family; trying to do right by each person; and trying to utilize the individual’s own capabilities [7].

#### 2.3.2. Japanese Version of the Approaches to Dementia Questionnaire (ADQ)

The ADQ was used to evaluate the awareness of person-centered care, and it has two subscales: hope and personhood. Higher scores indicate higher levels of person-centered care [10].

### 2.4. Ethical Considerations

The care staff were provided with an explanation regarding the purpose of the study. They provided informed consent before completing the questionnaire. Ethical approval for the study was granted by the Institutional Review Board of Hamamatsu University School of Medicine (No. 20-077).

### 2.5. Statistical Analysis

Exploratory factor analysis and confirmatory factor analysis were conducted to determine construct validity. Cronbach’s α was used to evaluate reliability. Correlations were calculated between the DL-DM, ADQ, and nurses’ attitudes toward the support for decision-making subscales to determine concurrent validity. Data were analyzed using IBM SPSS (v28.0.1.1., IBM, Armonk, NY, USA) and AMOS software (v26.0., IBM, Armonk, NY, USA).

## 3. Results

### 3.1. Participants’ Characteristics

The study comprised 138 participants, including 110 women (79.7%) and 28 men (20.3%). The mean age was 41.46 ± 2.3. Ninety-one (65.9%) participants were caregivers, and thirty-six (26.1%) were nurses (Table 1).

Table 2 provides the mean scores and standard deviations for the scales evaluating concurrent validity. The strongest endorsement was for “working as a team” with regard to attitudes toward support for decision-making by OwDs. The strongest endorsement was for “personhood” on the ADQ.

### 3.2. Validity Verification

#### 3.2.1. Exploratory Factor Analysis

The DL-DM was developed using exploratory factor analysis and covariance structure analysis. The exploratory factor analysis resulted in three factors (Table 3). Factor 1 was labeled “support for the formation and expression of intentions in daily life based on the life background and values of OwDs”. Factor 2 was labeled “Attitudes and devising ways to communicate regarding the formation and expression of intentions in the daily lives of OwDs”. Factor 3 was labeled “devising ways to support OwDs in realizing their intentions in daily life”. The highest factor loadings for the 12 items ranged from 0.385 to 0.827, explaining 61.04% of the cumulative contribution.

#### 3.2.2. Constructive Concept Validity

The confirmatory factor analysis goodness-of-fit indices in the final model were as follows: goodness of fit index (GFI) = 0.97, adjusted GFI (AGFI) = 0.91, comparative fit index (CFI) = 0.927, and root mean square error of approximation (RMSEA) = 0.048 (Figure 1). The indices indicate that the model fit was good.

#### 3.2.3. Coexistence Validity

Correlations evaluating concurrent validity are provided in Table 4. Generally, the three factors of the DL-DM correlated, as expected, with the nurses’ attitudes toward support for decision-making by OwDs subscales. Factor 1 was significantly correlated with three of the five subscales, and factors 2 and 3 were significantly correlated with all five subscales. With regard to the ADQ, the three factors of the DL-DM were significantly positively correlated with the personhood subscale, whereas only factor 1 was significantly correlated with the hope subscale.

### 3.3. Reliability Verification

#### 3.3.1. Cronbach’s Alpha Coefficient

The reliability analysis yielded a Cronbach’s α of 0.87 for the total scale.

#### 3.3.2. Item-Total Correlation Analysis

The correlation coefficients for all items and between the sum of the factor analysis results and each item were calculated.

#### 3.3.3. Test–Retest Reliability

Test–retest reliability was calculated after one week, resulting in a correlation coefficient of 0.856, which is considered significant.

## 4. Discussion

The purpose of this study was to develop and validate a scale to assess the daily-living decision-making support of care staff for OwDs in Japan. Approximately 80% of the participants were women. This may be due to the fact that the majority of caregivers and nurses in Japan are women. In this sample of care staff, approximately two-thirds of the participants had attended end-of-life care conferences. Approximately 40% had experience asking OwDs about ACP, and 30% had attended a decision-making workshop on ACP for OwDs. These characteristics of the participants indicate that care workers are involved in decision-making and ACP implementation and caregiving support, and that there is a high level of awareness and practice of ACP in Japanese facilities. In this regard, we developed the DL-DM to assess the daily-living decision-making support of the care staff for OwDs.

Factor analysis revealed that the DL-DM consisted of three factors: support for the formation and expression of intentions in daily life based on the life background and values of OwDs (Factor 1), attitudes and devising ways to communicate regarding the formation and expression of intentions in the daily lives of OwDs (Factor 2) and devising ways to support OwDs in realizing their intentions in daily life (Factor 3). These factors relate to the importance of decision-making support in terms of formation, expression, and realization of daily-life decisions based on the sense of values of OwDs. These three factors of formation, expression, and realization are consistent with the three components of the “decision-making support guidelines in the daily and social life of people with dementia [5]”.

Factor 1 includes four items that reflect support for verbal expression based on the patient’s background and values. The staff look for options that reflect how the OwD wants to live and be cared for, and their wishes and values with reference to life history and values (e.g., “at first we confirm the consistency of their remarks and the conventional sense of values in OwDs” and “we look for a choice based on their own intentions and values”). The importance of practicing care for the values and personal background of OwDs has been described from the perspective of person-centered dementia care [8]. In facilitating ACP, it is necessary to stay connected with OwDs and to allow for maximum participation in ACP in daily life [11]. If the person can speak but cannot express their true intentions, staff should first check to see if the person’s statements are consistent with their existing values. It is important for care staff to recognize and practice the values and life history of the OwD to support decision-making. In Europe and the United States, there are reports of shared decision-making, including ACP, in the care of OwDs with their proxies [12]. In Japanese nursing facilities, the care staff practices shared decision-making based on the person’s values and life background.

Factor 2 has four items that reflect actions taken by care staff to assist OwDs with communication (e.g., “we wait for the OwD to respond, even if their communication takes time, to draw the terms for their daily living” and “to form a will, we support them to make their choices independently and show concrete choices in their daily living according to their pace”). For ACP, individuals with moderate and severe dementia, especially OwDs, may have difficulty with verbal communication; thus, it is important to better understand their current quality of life, fears, and desires by responding to their emotions, paying attention to nonverbal communication, and observing their behavior [13]. It has also been reported that emotional control is important for decision-making in OwDs [14], and the communication of care preferences and assisting with communication correspond to Factors 1 and 2.

Factor 3 has four items that represent the need to support the person’s wishes in daily life care and the realization of the person’s will (e.g., “when OwDs experience difficulties in daily living, we provide them with ways and the care they expect” and “we collaborate and support them through inter-professional work, so that they can live independently and their ability is maximized”). Even when the person’s wishes cannot be fulfilled, support is provided to reconcile feelings by providing alternative means or by other means that reflect even a portion of the person’s wishes. The study also highlights that support for decision-making in daily life leads to decision-making at the end of life. Although the communication techniques and care methods used by care staff can facilitate ACP conversations with OwDs, decision support for OwDs in daily life has been neglected in dementia care practice. Older adults with severe dementia may express difficult intentions through facial expressions and behavior [15], but a person’s emotions can give direction to the decision-making process [8]. Therefore, it is important to interpret certain aspects of the OwD’s behavior and emotions to understand their intentions.

The covariance structure confirmed that the goodness-of-fit index was satisfied in the final model. Furthermore, in the evaluation of concurrent validity, Factors 1 and 2 were significantly positively correlated with the personhood subscale of the ADQ. These results indicate that decision-making support in daily living is associated with person-centered care [16]. The VIPS framework is a four-part definition of person-centered care, including valuing OwDs; an individualized care approach; understanding the OwD’s perspective; and a supportive social environment for OwDs [16]. The DL-DM specifically addresses the formation and expression of intentions in daily life, attitudes, and communication from the perspective of OwDs. This suggests that decision-making support in daily living is based on person-centered care. Factor 3, meanwhile, emphasizes the realization of daily-living decisions. Although all decisions have not been realized and implemented, these care plans related to daily-living decisions are indirectly handled considering OwD’s will, tp ensure they are as close as possible to the person’s wishes.

In addition, the DL-DM factors correlated with the five factors of the nurses’ attitudes toward support for decision-making by OwDs. The DL-DM demonstrated validity and reliability as a measure of care-staff support of decision-making in daily life for OwDs. The association between daily decision-making support and person-centered care for OwDs suggests that the DL-DM can provide useful information about OwDs’ everyday decision-making and facilitate understanding of the relationship between care-staff involvement in decision-making and the well-being of OwDs.

The ADQ measures person-centered care. However, using the DL-DM, care staff can measure their current ACP competence. Furthermore, it can be used to evaluate the effectiveness of ACP development training for care staff.

Decision support should be based on person-centered care and enable OwDs to live better and make decisions in their daily lives, especially at the end of life, as well as support OwDs’ choice of medical care at the end of life. However, it should be noted that the study had several limitations that need to be addressed in future research. For example, one major limitation is generalizability, the study included only a small number of care staff, in particular, nurses, and from only two facilities. Therefore, further research is needed, with larger sample sizes across more geriatric care facilities.

## 5. Conclusions

The DL-DM demonstrated validity and reliability as a measure to assess decision-making support by care staff in the daily lives of OwDs, suggesting an association between daily decision-making support and person-centered care for OwDs. The DL-DM provides useful information about decision-making in the daily lives of OwDs and is a useful tool for understanding the relationship between the decision-making involvement of care staff and the well-being of OwDs. Decision-making for daily-living support should be based on person-centered care and enable OwDs to live a better quality of life and make decisions about their daily lives. It should also support medical choices, especially at the end of life. The DL-DM can promote the future efforts of care staff regarding ACP for OwDs.

## Figures and Tables

**Figure 1 ijerph-19-13664-f001:**
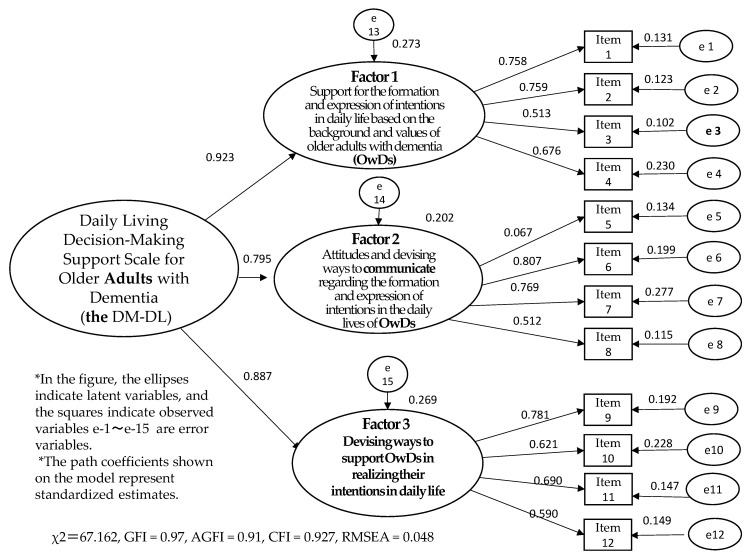
Covariance structure analysis of the Daily Living Decision-Making Support Scale for Older Adults with Dementia (DL-DM).

**Table 1 ijerph-19-13664-t001:** Participants’ characteristics.

Items		Mean	SD
Age		41.5	12.3
Years of experience in geriatric care		13.5	9.4
**Items**		**n**	**%**
Gender	Men	28	20.3
	Women	110	79.7
Type of Job	Caregiver	91	65.9
	Nurse	36	26.1
	OT	9	6.5
	PT	2	1.4
Experienced end of life care conference	Yes	89	64.5
	No	49	35.5
Experienced asking OwDs on advanced care planning	Yes	61	44.2
	No	77	55.8
Participated in decision making and advanced care planning workshop	Yes	41	29.7
	No	97	70.3
Participated in workshop on person-centered care	Yes	123	89.1
	No	15	10.9

OT, occupational therapist; PT, physical therapist; OwDs, Older adults with dementia.

**Table 2 ijerph-19-13664-t002:** Means and standard deviations (SD) of evaluation scales.

Evaluation Scale	Mean	SD
Nurses’ attitudes toward support for decision-making by OwDs		
1. Working as a team	16.15	2.68
2. Respecting the individual’s wishes	14.76	2.82
3. Trying to understand the family	9.41	1.80
4. Trying to do right by each person	8.69	1.87
5. Trying to utilize the individual’s own capabilities	9.38	1.62
Approaches to Dementia Questionnaire		
1. Hope	28.64	3.92
2. Personhood	44.33	4.22

OwDs, older adults with dementia.

**Table 3 ijerph-19-13664-t003:** Factor analysis of the Daily Living Decision-Making Support Scale for Older Adults with Dementia.

The Daily Living Decision-Making Support Scale for Older Adults with Dementia	Factor 1	Factor 2	Factor 3	Mean	SD	I-T Correlation
1. Support for the formation and expression of intentions in daily life based on the life background and values of people with dementia (OwDs)						
(1) In reference to the past history of life and sense of values of OwDs, we look for a choice based on their own intentions and values reflected in their desire for life and care.	**0.713**	0.287	0.212	3.20	0.53	0.702 **
(2) Although OwDs can talk, they cannot express their real intention sufficiently (especially those with a moderate degree of dementia). Therefore, at first we confirm the consistency of their remarks and the conventional sense of values in OwDs.	**0.682**	0.354	0.15	3.09	0.68	0.725 **
(3) During daily living, we explain using simple terms, letters, and real things so that OwDs can express their own will.	**0.465**	0.123	0.221	3.39	0.56	0.519 **
(4) When it is difficult for OwDs to express their choices and intentions through verbal communication, we consider their will from their facial expressions and behavior.	**0.449**	0.355	0.300	3.36	0.52	0.673 **
2. Attitudes and devising ways to communicate regarding the formation and expression of a will in the daily lives of OwDs						
(5) We wait for the OwD to respond, even if their communication takes time, to draw the terms for their daily living, so that they can freely express their own will through behavior and facial expressions.	0.127	**0.775**	0.165	2.96	0.66	0.636 **
(6) We accept the thoughts of the OwD at any time to show support for their concrete choices in daily living, depending on the pace of the person’s intentions.	0.413	**0.636**	0.209	3.09	0.67	0.737 **
(7) To form a will, we support them to make their choices independently and show concrete choices in their daily living according to their pace. We also listen to and acknowledge their expressions.	0.402	**0.594**	0.229	3.11	0.70	0.730 **
(8) When it is difficult for OwDs to express their choices and intentions through verbal communication, we speak on their behalf and examine their sense of values while taking note of the consistencies.	0.262	**0.385**	0.168	2.96	0.62	0.565 **
3. Devising ways to support OwDs in realizing their intentions in daily life.						
(9) When OwDs experience difficulties in daily living, we provide them with ways and the care they expect. We plan emotional stability for OwDs.	0.173	0.164	**0.827**	3.26	0.61	0.629 **
(10) We accept the decisions of OwDs. If a person declines care, we respect it without forcing care on them.	0.133	0.143	**0.612**	3.33	0.67	0.551 **
(11) If the will of OwDs are not granted, we prepare substitute care and support to reconcile at least part of their requests and wishes.	0.422	0.209	**0.475**	3.16	0.62	0.673 **
(12) When OwDs have mutual understanding and daily living, we collaborate and support them through inter-professional work, so that they can live independently and their ability is maximized.	0.254	0.197	**0.462**	3.29	0.64	0.468 **
Eigenvalue	5.103	1.266	0.955	□	□	□
Factor contribution ratio	42.527	10.553	7.959	□	□	□
Accumulation contribution ratio	61.038			□	□	□
Cronbach’s α	0.826	0.786	0.753	□	□	□
Cronbach’s α for the 12 items	□	□	0.873	□	□	□

Factor sampling (the main factor analysis) rolling method: Balimax method KMO level = 0.871 *p* < 0.0001 with the normalization of Kaiser. I-T: Item-total. ** *p* < 0.01.

**Table 4 ijerph-19-13664-t004:** Correlations between the Daily Living Decision-Making Support Scale for Older Adults with Dementia and the evaluation scales.

Daily Living Decision-Making Support Scale for Older Adults with Dementia		Nurses’ Attitudes toward Support for Decision-Making by OwDs						Approaches to Dementia Questionnaire	
		Correlation Coefficient	Working as a Team	Respecting the Individual’s Wishes	Trying to Understand the Family	Trying to Do Right by Each Person	Trying to Utilize the Individual’s Own Capabilities	Hope	Personhood
Factor 1	Support for the formation and expression of intentions in daily life based on the life background and values of OwDs	r	0.246	0.162	0.210	0.111	0.318	0.212	0.237
		*p*-value	0.004	0.057	0.013	0.193	0.000	0.013	0.005
Factor 2	Attitudes and devising ways to communicate regarding the formation and expression of intentions in the daily lives of OwDs	r	0.360	0.391	0.426	0.343	0.416	0.119	0.290
		*p*-value	0.000	0.000	0.000	0.000	0.000	0.165	0.000
Factor 3	Devising ways to support OwDs in realizing their intentions in daily life	r	0.367	0.297	0.300	0.236	0.359	0.024	0.249
		*p*-value	0.000	0.000	0.000	0.005	0.000	0.780	0.005

OwDs, older adults with dementia.

## Data Availability

The data that support the findings of this study are available from the corresponding author upon reasonable request.

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
