# Peer review of "Care Staff’s Daily Living Decision-Making Support Scale for Older Adults with Dementia in Japan: Development of Validity and Reliability"

_ijerph, 2022, doi:10.3390/ijerph192013664_

Round 1

Reviewer 1 Report

“Daily Living Decision-Making Support Scale for Older Adults with Dementia in Japan: Development on Validity and Reliability” by Suzuki and colleagues is a study to develop and validate a scale to assess decision support in daily life for caregivers of elderly persons with dementia in Japan.

Although the topic is interesting and of considerable importance in the scientific literature, the manuscript needs major revisions before publication in IJERPH.

Abstract. This section is of paramount importance because it should entice the reader to read the work in full. In this case, although the authors have described the content of their manuscript in detail, it is in my opinion incomplete. In fact, it would be appropriate to concisely add a background that well introduces the topic addressed in the manuscript. Furthermore, the abstract lacks a concluding sentence suggesting possible future perspectives that might emerge from the results obtained.

Introduction. This section should be significantly deepened and provide a broader perspective on the topic addressed, with particular reference to the definition of dementia, prevalence and associated socio-economic and health impact. Furthermore, the authors could include examples of associated diseases that involve progressive and irreversible degeneration of the central nervous system and impair cognitive functions. In this regard, the recent review by Diociaiuti et al. summarises the topic in detail (Diociaiuti M, Bonanni R, Cariati I, Frank C, D'Arcangelo G. Amyloid Prefibrillar Oligomers: The Surprising Commonalities in Their Structure and Activity. Int J Mol Sci. 2021 Jun 16;22(12):6435. doi: 10.3390/ijms22126435. PMID: 34208561; PMCID: PMC8235680.).

Materials and Methods. Section 2.2 – Although the characteristics of the study participants were described in the results section, the authors should at least include the number of participants and the average age in this section. Furthermore, they should specify whether the recruitment of participants was done according to inclusion and exclusion criteria.

Results. Section 3.1 – The authors report that they included 110 women and 28 men in the study. Why is there such a large discrepancy between the enrolled participants? It would be appropriate to argue this.

Tables and Figures. Was a statistical analysis conducted for the variables considered? The authors report values in Tables 1 and 2 as mean ± SD, but there is no reference to statistical significance. Please conduct a statistical analysis and report the p-values in the tables. This should also be reported in the text. Furthermore, the authors should make an effort to better summarise the concepts in Table 3, as it is not immediately comprehensible to the reader. Same comment for Figure 1.

Discussion. This section should summarise the results obtained from the study, discussing and critically arguing them. In addition, the authors should expand the bibliography, comparing their results with data already published in the literature. For example, reference could also be made to the impact of exercise on the cognitive functions of individuals with dementia. Several scientific evidences have addressed this topic (Lamb SE, Sheehan B, Atherton N, Nichols V, Collins H, Mistry D, Dosanjh S, Slowther AM, Khan I, Petrou S, Lall R; DAPA Trial Investigators. Dementia And Physical Activity (DAPA) trial of moderate to high intensity exercise training for people with dementia: randomised controlled trial. BMJ. 2018 May 16;361:k1675. doi: 10.1136/bmj.k1675. PMID: 29769247; PMCID: PMC5953238.; Vizzi L, Padua E, D'Amico AG, Tancredi V, D'Arcangelo G, Cariati I, Scimeca M, Maugeri G, D'Agata V, Montorsi M. Beneficial Effects of Physical Activity on Subjects with Neurodegenerative Disease. J Funct Morphol Kinesiol. 2020 Dec 16;5(4):94. doi: 10.3390/jfmk5040094. PMID: 33467309; PMCID: PMC7804865.), as well as the importance of an adequate diet (Dominguez LJ, Veronese N, Vernuccio L, Catanese G, Inzerillo F, Salemi G, Barbagallo M. Nutrition, Physical Activity, and Other Lifestyle Factors in the Prevention of Cognitive Decline and Dementia. Nutrients. 2021 Nov 15;13(11):4080. doi: 10.3390/nu13114080. PMID: 34836334; PMCID: PMC8624903.). Therefore, a considerable reworking of this section is recommended.

Conclusions. It is important that this section explicitly states the added value of this study and what innovation it provides to the current literature.

Author Response

We appreciate the time and effort that you and the reviewers have dedicated to reviewing our manuscript and providing valuable and informative feedback. We are grateful to the reviewers for their insightful comments on our paper.

Reviewer Comments

Introduction. This section should be significantly deepened and provide a broader perspective on the topic addressed, with particular reference to the definition of dementia, prevalence and associated socio-economic and health impact. Furthermore, the authors could include examples of associated diseases that involve progressive and irreversible degeneration of the central nervous system and impair cognitive functions. In this regard, the recent review by Diociaiuti et al. summarises the topic in detail (Diociaiuti M, Bonanni R, Cariati I, Frank C, D'Arcangelo G. Amyloid Prefibrillar Oligomers: The Surprising Commonalities in Their Structure and Activity. Int J Mol Sci. 2021 Jun 16;22(12):6435. doi: 10.3390/ijms22126435. PMID: 34208561; PMCID: PMC8235680.).

⇒Response: As you pointed out, we have added details regarding the definition of dementia, prevalence rates, and health status.

“The World Health Organization (WHO) defined dementia as “a syndrome—usually of a chronic or progressive nature—that leads to deterioration in cognitive function beyond what affects memory, thinking, orientation, comprehension, calculation, learning capacity, language, and judgement“ [1]. Dementia not only causes cognitive decline but also impairs physical function, leaving older adults in need of nursing care or bedridden.”

World Health Organization. Dementia.

https://www.who.int/news-room/fact-sheets/detail/dementia

The current study deals with advanced care planning (ACP) in dementia. Although exercise and diet are important in relation to dementia, they fall outside the scope of this study. If we were to discuss diet and exercise, it would greatly increase the size of the manuscript. Thus, we did not include them in the manuscript.

Reviewer Comments

Materials and Methods. Section 2.2 – Although the characteristics of the study participants were described in the results section, the authors should at least include the number of participants and the average age in this section. Furthermore, they should specify whether the recruitment of participants was done according to inclusion and exclusion criteria.

⇒Response: As per your suggestion, I have included information regarding the number of participants and their average age in the Methods section. I have mentioned the inclusion and exclusion criteria as well.

“The inclusion criterion for the participants was working as care staff in geriatric care facilities, while the exclusion criteria were not being able to give consent, working part-time, and being on maternity leave or other extended leaves of absence. This study comprised 138 participants, and the mean age was 41.46±2.3.”

Reviewer Comments

The authors report that they included 110 women and 28 men in the study. Why is there such a large discrepancy between the enrolled participants? It would be appropriate to argue this.

⇒Response: Thank you for pointing this out. We have added more details regarding this as follows:

“Approximately 80% of the participants were women. This may be due to the fact that the majority of caregivers and nurses in Japan are women.”

Reviewer Comments

Discussion. This section should summarise the results obtained from the study, discussing and critically arguing them. In addition, the authors should expand the bibliography, comparing their results with data already published in the literature. For example, reference could also be made to the impact of exercise on the cognitive functions of individuals with dementia. Several scientific evidences have addressed this topic (Lamb SE, Sheehan B, Atherton N, Nichols V, Collins H, Mistry D, Dosanjh S, Slowther AM, Khan I, Petrou S, Lall R; DAPA Trial Investigators. Dementia And Physical Activity (DAPA) trial of moderate to high intensity exercise training for people with dementia: randomised controlled trial. BMJ. 2018 May 16;361:k1675. doi: 10.1136/bmj.k1675. PMID: 29769247; PMCID: PMC5953238.; Vizzi L, Padua E, D'Amico AG, Tancredi V, D'Arcangelo G, Cariati I, Scimeca M, Maugeri G, D'Agata V, Montorsi M. Beneficial Effects of Physical Activity on Subjects with Neurodegenerative Disease. J Funct Morphol Kinesiol. 2020 Dec 16;5(4):94. doi: 10.3390/jfmk5040094. PMID: 33467309; PMCID: PMC7804865.), as well as the importance of an adequate diet (Dominguez LJ, Veronese N, Vernuccio L, Catanese G, Inzerillo F, Salemi G, Barbagallo M. Nutrition, Physical Activity, and Other Lifestyle Factors in the Prevention of Cognitive Decline and Dementia. Nutrients. 2021 Nov 15;13(11):4080. doi: 10.3390/nu13114080. PMID: 34836334; PMCID: PMC8624903.). Therefore, a considerable reworking of this section is recommended.

⇒Response: Thank you for your valuable input.

This study aimed to develop a scale on ACP for older adults with dementia. Given that we did not explore exercise as a factor in relation to dementia, we cannot present related results and comment on the relationship between cognitive functions of individuals with dementia and exercise; thus, we have not discussed this in the manuscript.

Reviewer Comments

Conclusions. It is important that this section explicitly states the added value of this study and what innovation it provides to the current literature.

⇒Response: Thank you for your comment. We have added the following sentence to the Conclusion:

“The DL-DM can promote the future efforts of care staff regarding ACP for OwDs.”

Reviewer 2 Report

Thank you for allowing me to review this article.  The authors have developed and validated a scale to assess daily-decision making support efforts of care staff caring for OwDs.  The manuscript was well laid out, and the science and results seemed sound.  I do have a few comments that I believe would strengthen the manuscript:

1) When reading the title, it is not clear that the population this scale is meant for is the care staff of those with dementia.  I originally thought it was designed to be given to the patients.  Please update the title so that this is clear to the readers.

2) What kind of sample was the focus group completed with (i.e., nurses, occupational therapists, OwDs, etc.)?

3) Please provide details of the qualitative analysis that was completed.  Was there a specific protocol followed?  What kind of software was used?

Author Response

We appreciate the time and effort that you and the reviewers have dedicated to reviewing our manuscript and providing valuable and informative feedback. We are grateful to the reviewers for their insightful comments on our paper.

Reviewer Comments

1) When reading the title, it is not clear that the population this scale is meant for is the care staff of those with dementia.  I originally thought it was designed to be given to the patients.  Please update the title so that this is clear to the readers.

⇒Response: Thank you for pointing this out. We have revised the title as follows:

“Care Staff’s Daily Living Decision-Making Support Scale for Older Adults with Dementia in Japan: Development of Validity and Reliability”

Reviewer Comments

2) What kind of sample was the focus group completed with (i.e., nurses, occupational therapists, OwDs, etc.)?

⇒Response: Addressing you concern, we have added the following details regarding the focus group:

“Focus group interviews were conducted with care staff, such as caregivers, nurses, physical therapists and occupational therapists, at a geriatric care facility. Results were analyzed using a qualitative analysis method called content analysis.”

3) Please provide details of the qualitative analysis that was completed.  Was there a specific protocol followed?  What kind of software was used?

⇒ “Results were analyzed using a qualitative analysis method called content analysis.”

Reviewer 3 Report

This study is of great clinical significance because it developed and validated a scale to assess the daily life decision-making support of caregiving staff for elderly people with dementia in Japan. Manuscript is well written and consistent.

How was the sample size for this study calculated?

Author Response

We appreciate the time and effort that you and the reviewers have dedicated to reviewing our manuscript and providing valuable and informative feedback. We are grateful to the reviewers for their insightful comments on our paper.

Reviewer Comments

How was the sample size for this study calculated?

⇒Response: We have added details regarding the calculation of sample size in the Methods section as follows:

“According to Floyd, for a sample size of 100 participants, five participants per variable would be considered adequate to yield reliable results, while 10 participants per variable would be sufficient for a sample size of less than 100 participants [9]. Thus, a sample size of 138 participants in this study was sufficient for factor analysis and covariance structure analysis.”

Floyd, F.J.; Widaman, K.F. Factor analysis in the development and refinement of clinical assessment instruments. Psychol Assess 1995, 7(3), 286–299. doi: 10.1037/1040-3590.7.3.286.

Reviewer 4 Report

The authors conducted a questionnaire survey of 138 nurses from two geriatric health care institutions and developed a scale to assess the daily living decision-making support of care staff for older adults with dementia in Japan. However, I have some concerns about this manuscript. 

Major comments

1. The sample size of this study is small. Hope to supplement more data.

2. Although the author has fully explained and illustrated the reliability of this study, there are still some evaluations that are expected to be presented in the article: test-retest reliability, inter-rater reliability, split-half reliability, etc. Please complete this part.

3. "DL-DM are provided on a 4-point Likert scale" is mentioned in the article. Does the author have a reasonable interpretation of the scale score? Can the decision-making ability of care staff be quantitatively evaluated?

4. Please add the evaluation of content validity and relevant statistical results.

5. Is the scale completed by the participants themselves? Please add to this point in the text. If rater assistance is required, please supplement the statistical data such as inter-rater reliability.

Minor comments

6. The inclusion and exclusion criteria of participants should be included.

7. Please enumerate the results of the reliability and validity, and divide them into paragraphs.

8. The layout of Table 2 is not concise enough. It is recommended that items be clear and prioritized. For example, assign numbers to subscales.

9. What are the advantages of DL-DM from results, over ADQ and Nurses' attitudes toward support for decision-making by OwDs? Please provide additional clarification in the article.

10. The study used survey data from only two geriatric care facilities. Were the findings influenced by the level of "nursing staff training in intervention or routine care"? In addition to geriatric care facilities, could the author apply the DL-DM scale of care staff to other types of care, such as the hospital staff, family caregivers, or family members?

11. The concept of "person-centered care" is mentioned in the article, to assess the Daily Living decision-making support of care staff for older adults with dementia. Could you provide additional scale data (related to patient experience,  improvement of symptoms, and quality of life) to support the correlation between the evaluation of DL-DM in the questionnaire and the decision-making ability of care staff? Is there evidence that higher scores for caregivers lead to a better quality of life for OwDs and the ability to make decisions about OwDs’ daily lives?

Author Response

We appreciate the time and effort that you and the reviewers have dedicated to reviewing our manuscript and providing valuable and informative feedback. We are grateful to the reviewers for their insightful comments on our paper.

  1. The sample size of this study is small. Hope to supplement more data.

⇒Response: Thank you for your comment. We have provided an explanation regarding the sample size of the study as follows:

“According to Floyd, for a sample size of 100 participants, five participants per variable would be considered adequate to yield reliable results, while 10 participants per variable would be sufficient for a sample size of less than 100 participants [9]. Thus, a sample size of 138 participants in this study was sufficient for factor analysis and covariance structure analysis.”

Floyd, F.J.; Widaman, K.F. Factor analysis in the development and refinement of clinical assessment instruments. Psychol Assess 1995, 7(3), 286–299. doi: 10.1037/1040-3590.7.3.286.

Reviewer Comments

  1. Although the author has fully explained and illustrated the reliability of this study, there are still some evaluations that are expected to be presented in the article: test-retest reliability, inter-rater reliability, split-half reliability, etc. Please complete this part.

⇒Response: We have added information regarding test-retest reliability in the revised manuscript as follows:

“Test-retest reliability was calculated after one week, resulting in a correlation coefficient of .0856, which is considered significant.”

Reviewer Comments

  1. "DL-DM are provided on a 4-point Likert scale" is mentioned in the article. Does the author have a reasonable interpretation of the scale score? Can the decision-making ability of care staff be quantitatively evaluated?

⇒Response: Japanese tend to choose options closest to the center as their responses, and as a 4-point Likert scale distributes them evenly, it was considered suitable to quantitatively assess the decision-making ability of caregivers toward older adults with dementia.

Reviewer Comment

  1. Please add the evaluation of content validity and relevant statistical results.

  1. Is the scale completed by the participants themselves? Please add to this point in the text. If rater assistance is required, please supplement the statistical data such as inter-rater reliability.

⇒Response: In the Results section, we have added statistics on reliability, and the reliability and validity of the results were itemized for clarity.

Minor comments

Reviewer Comment

  1. The inclusion and exclusion criteria of participants should be included.

⇒Response: Thank you for pointing this out. The inclusion and exclusion criteria have been noted in the Methods section under ‘2.2 Participants’.

“The inclusion criterion for the participants was working as care staff in geriatric care facilities, while the exclusion criteria were not being able to give consent, working part-time, and being on maternity leave or other extended leaves of absence.”

.

Reviewer Comment

  1. Please enumerate the results of the reliability and validity, and divide them into paragraphs.

⇒Response: In the Results section, the results of reliability and validity were indicated and segregated into paragraphs.

Reviewer Comment

  1. The layout of Table 2 is not concise enough. It is recommended that items be clear and prioritized. For example, assign numbers to subscales.

⇒Response: thank you for your suggestion. We have numbered the subscales in Table 2.

Reviewer Comment

  1. What are the advantages of DL-DM from results, over ADQ and Nurses' attitudes toward support for decision-making by OwDs? Please provide additional clarification in the article.

⇒Response: We have included the following statements in the Discussion section:

“The ADQ measures person-centered care. However, using the DL-DM, care staff can measure their current ACP competence. Furthermore, it can be used to evaluate the effectiveness of ACP development training for care staff.”

Reviewer Comment

  1. The study used survey data from only two geriatric care facilities. Were the findings influenced by the level of "nursing staff training in intervention or routine care"? In addition to geriatric care facilities, could the author apply the DL-DM scale of care staff to other types of care, such as the hospital staff, family caregivers, or family members?

⇒Response: We would like to explore the possibility of applying the DL-DM

Reviewer Comment

  1. The concept of "person-centered care" is mentioned in the article, to assess the Daily Living decision-making support of care staff for older adults with dementia. Could you provide additional scale data (related to patient experience, improvement of symptoms, and quality of life) to support the correlation between the evaluation of DL-DM in the questionnaire and the decision-making ability of care staff?

⇒Response: Unfortunately, we do not have any data related to patient experience,  improvement of symptoms, and quality of life that can be added to the manuscript.

Reviewer Comment

 Is there evidence that higher scores for caregivers lead to a better quality of life for OwDs and the ability to make decisions about OwDs’ daily lives?

⇒Response: The ADQ measures person-centered care, while nurses’ attitudes toward support for decision-making by OwDs measures attitudes toward support for decision-making. In the examination of coexistence validity, a significant association of the DL-DM with the ADQ and Nurses’ attitudes toward support for decision-making by OwDs was observed, which led us to believe that this scale could measure the ability of care staff to support decision making.

Round 2

Reviewer 1 Report

The authors partly fulfilled my requests. I think the introduction could be further improved and enriched with various information from the literature. However, the choice is the authors'. 

Author Response

We appreciate the time and effort that you and the reviewers have dedicated to reviewing our manuscript and providing valuable and informative feedback.

Reviewer 1

Reviewer Comments

The authors partly fulfilled my requests. I think the introduction could be further improved and enriched with various information from the literature. However, the choice is the authors'.

⇒Response: Thank you for your kind comments. As you pointed out, we need the introduction could be enriched with various information. In the introduction to this study, I would like to request only information on ACP.

We are grateful to the reviewers for their insightful comments on our paper.